molecular biology/ecology/genetics

Cepe de Bordeaux, *Boletus edulis*, sporocarp, genetic diversity, relatedness, population structure

**Author for correspondence:**
J. I. Hoffman
e-mail: joseph.hoffman@uni-bielefeld.de

# Genetic analysis of *Boletus edulis* suggests that intra-specific competition may reduce local genetic diversity as a woodland ages

J. I. Hoffman[1,2], R. Nagel[1], V. Litzke[1], D. A. Wells[1,3,4] and W. Amos[3]

[1]Department of Animal Behaviour, Bielefeld University, 33501 Bielefeld, Germany
[2]British Antarctic Survey, High Cross, Madingley Road, Cambridge CB3 0ET, UK
[3]Department of Zoology, University of Cambridge, Downing Street, Cambridge CB2 3EJ, UK
[4]School of Natural Science and Psychology, Liverpool John Moores University, Liverpool, UK

JIH, 0000-0001-5895-8949; RN, 0000-0002-2925-1028;
VL, 0000-0002-3170-6060; WA, 0000-0002-0971-9914

Ectomycorrhizal fungi are key players in terrestrial ecosystems yet their mating systems and population dynamics remain poorly understood. We investigated the fine-scale relatedness structure and genetic diversity of *Boletus edulis*, one of the world's most commercially important wild mushrooms. Microsatellite genotyping of fruiting bodies from 14 different sites around Bielefeld in Germany revealed little in the way of population structure over a geographic scale of several kilometres. However, on a more local scale we found evidence for elevated relatedness as well as inbreeding. We also observed a significant negative association between the genetic diversity of fruit and the age of the trees under which they were sampled. Taken together, our results suggest that as genets mature, they compete and potentially create conditions under which further spores struggle to become established. By implication, even though this species is widely picked, propagules remain common enough to create strong competition when new habitats become available.

## 1. Introduction

Ectomycorrhizal fungi (ECM) are important components of almost all terrestrial ecosystems, being associated with and aiding the growth of many dominant tree species in temporal and boreal forests [1]. ECMs also play crucial roles in nutrient recycling,

breaking down complex biochemicals that would otherwise act as nutrient sinks and contributing towards the global carbon cycle [2]. Given their ecological importance, there is a pressing need to better understand the population dynamics and life histories of ECM fungi, since this will inform about a critical component of how ecosystems function and persist [3]. In particular, studies of the temporal and spatial distributions of ECM fungi, as well as of the origin and maintenance of their genetic variation, are essential for understanding fundamental life-history processes that shape ECM populations and communities [4].

Despite their importance, relatively little is known about the population structure, genetic diversity and mating systems of ECM fungi. Unique genetic individuals, termed 'genets' [5] arise from sexual reproduction via basidospores and then expand 'vegetatively' as mycelia (the aggregate of filamentous growth) in conjunction with their host(s). However, many aspects of the ecology of ECM fungi remain poorly understood, including patterns of dispersal, genet longevity and the importance of local competition and environmental heterogeneity to establishment and persistence [6]. Several molecular genetic studies have mapped and analysed sporocarps (fruiting bodies) as these can be conveniently sampled [4,7–14] but few are extensive enough over space and/or time to build up a complete picture. Moreover, it is only recently that population genetic tools such as microsatellites have been developed with the resolution to identify individual genets [9,10,15,16].

Most ECM fungal spores are thought to be deposited very close to the sporocarp that produced them [17–19]. However, of those that are not, many possible mechanisms exist by which effective dispersal may occur. Being light, a proportion of spores will usually be blown away by wind, while animals may also be important vectors [20]. Many sporocarps are eaten by mammals such as deer and wild boar, and there is evidence that spores can persist in the gut long enough to remain viable after defaecation [21]. The majority of sporocarps are also visited by flies belonging to the family *Mycetophilidae* that lay eggs at the base of the stem into which the resulting larvae move to feed [22]. During egg-laying, these flies would probably become covered in spores, raising the possibility that flies may act as more targeted vectors. Exactly which forms of dispersal dominate will have a major impact on the resulting population structure. If most spores geminate near the sporocarp that produced them, fine-scale population structuring should result, but this is rarely found. For example, *Russula brevipes* appears unstructured over geographical scales of up to a kilometre, despite strong population structure being present over scales of thousands of kilometres [3]. Such a pattern would be consistent with population genetic theory, which suggests that low levels of gene flow are usually enough to prevent population differentiation. Consequently, even if the proportion of spores that end up being transported tens or hundreds of kilometres by wind, either directly or on the bodies of insects, is apparently negligible, it might be enough to cause approximate panmixia.

At a local level, population structure should also reflect a balance between the proportion of spores that establish mycelia, the longevity of the mycelia and competition among different genets. It has been suggested that new stands of trees are colonized by large numbers of propagules and that the resulting genets compete with each other, leading to some either being prevented from fruiting, perhaps diverting resources to vegetative reproduction, or even dying [4,8–10]. As a result, woodland age and genet diversity may be inversely related.

*Boletus edulis*, known variously as the penny bun, cèpe de Bordeaux, porcino or Steinpilz (figure 1) is arguably the most important wild commercially harvested mushroom [23–25] and offers an excellent system with which to investigate fungal population structure. There have also been several attempts to cultivate this mushroom [26] and understanding its population structure may facilitate this process. *B. edulis* forms part of a monophyletic group that evolved around 40–50 Ma [27–29], probably in tropical Asia [30]. This species has a broad geographic distribution across Eurasia and North America and has also been introduced into several Southern Hemisphere countries including South Africa and New Zealand [23]. Its hosts include pine, birch, beech, oak, spruce, chestnut and, above the treeline, rock rose, raising questions about whether local adaptation occurs to whichever host is locally dominant. Furthermore, where *B. edulis* occurs it is often abundant, allowing systematic sampling of sporocarps to be conducted over multiple spatial scales and through time.

Here, we used *B. edulis* samples collected from around Bielefeld in northwest Germany and genotyped for a panel of seven microsatellites to explore the relationship between genet diversity and host tree stand age. We report evidence that genet diversity is indeed highest among younger stands of trees and declines with increasing woodland age.

## 2. Results

In order to investigate patterns of fine-scale population structure in *B. edulis*, we conducted near-exhaustive fine-scale sampling of sporocarps from 14 sites around Bielefeld in Germany (figure 2). We

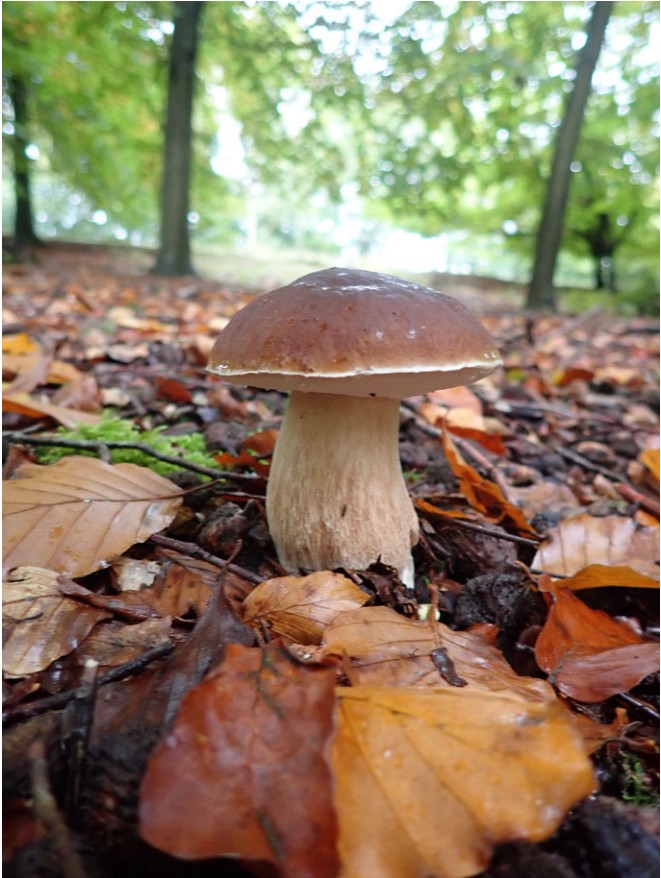

**Figure 1.** Fruiting body of *Boletus edulis* in beech woodland. Photocredit: Bill Amos.

then developed a panel of seven microsatellites (electronic supplementary material, table S1), which we genotyped in a total of 134 sporocarps (table 1). These loci carried on average 6.6 alleles (range = 2–11, electronic supplementary material, table S1) and none deviated significantly from Hardy–Weinberg equilibrium in any of the populations after false discovery rate correction (electronic supplementary material, table S2). The probability of identity across all loci after removing clonal replicates from each population was low at $3.3 \times 10^{-6}$.

## 2.1. Patterns of genet abundance

The sampling sites showed considerable variation both in the total number of sporocarps produced and in the number of genets (figure 3 and table 1). However, genet number was unrelated to the number of sporocarps (Pearson's $r = 0.508$, $p = 0.064$). For example, only six genets were discovered in a sample of 38 sporocarps from site one, whereas a smaller sample of 20 sporocarps from site two yielded 14 genets (figure 3). Many genets only produced sporocarps on a single occasion, whereas others fruited at multiple time points and some were productive over almost the entire duration of the season (electronic supplementary material, figure S1). However, interpretation of the numbers of sporocarps has to be conducted with caution. *B. edulis* is attractive both to human foragers and to many animals and while both may leave behind evidence that material has been eaten or removed, this does not have to be the case. Consequently, some unknown proportion will have been overlooked. Here, we assume that unsampled sporocarps represent a random subset with respect to genotype while at the same time acknowledging that there may be some dependency on fruiting intensity: larger flushes are likely to be more visible both to us and to other humans and animals.

## 2.2. Fine-scale relatedness structure

After clone-correcting the full genetic dataset to produce a restricted dataset of 54 unique genets, false discovery rate (FDR)-corrected pairwise $F_{st}$ values among the sample sites were mostly not statistically significant (electronic supplementary material, table S3). The only exceptions were comparisons

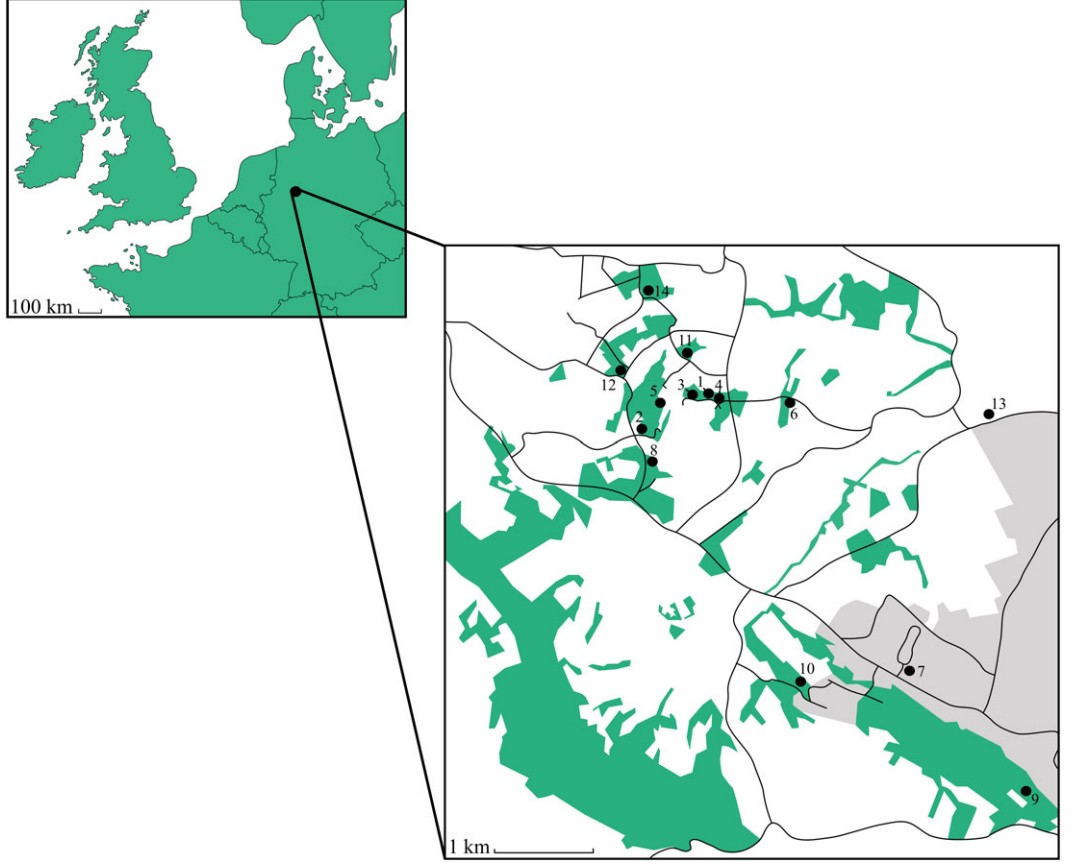

**Figure 2.** A map of *B. edulis* sampling locations around Bielefeld in Germany. Sporocarps were collected from a total of 14 sample sites, which are numbered in descending order of sporocarp and genet abundance. Forest patches are shown in green and urban areas are depicted in grey.

involving the locality with the highest genetic diversity (site two) and sites one and four. However, spatial autocorrelation analysis of *B. edulis* genets (figure 4) showed a strong and significant positive autocorrelation within the smallest distance class (250 m), suggesting that levels of relatedness are higher on average within than among sample sites. Significantly negative *r* values were also obtained for the 500, 750, 2250 and 3750 m distance classes. However, the permutation and bootstrap estimates of *r* were relatively imprecise for the majority of distance classes above 1000 m due to rapidly diminishing numbers of pairwise comparisons in the autocorrelation analysis.

To further investigate fine-scale patterns of relatedness, we generated a matrix of pairwise Lynch and Ritland relatedness (*r*) values among all of the genets. Relatedness varied significantly among the sample sites (Kruskal–Wallis test; $\chi^2 = 41.352$, d.f. = 13, $p < 0.001$). Plotting pairwise relatedness values as a heatmap revealed a striking pattern in which clusters of highly related individuals were readily apparent within some but not all of the sites (figure 5). Specifically, close relatives were common at a number of localities, most obviously site two but also to a lesser extent sites seven to ten.

## 2.3. Inbreeding

High levels of local relatedness among sexually reproducing genets might be expected to be reflected in elevated levels of inbreeding. To investigate this, we quantified the variance in inbreeding among *B. edulis* genets by calculating the two-locus disequilibrium ($g_2$) for our dataset. The resulting estimate of 0.055 was high relative to many species but the 95% confidence interval was broad and overlapped zero (−0.042 to 0.168, $p = 0.077$). Nevertheless, significant differences in individual standardized multilocus heterozygosity (sMLH) were found among the study sites (Kruskal–Wallis test; $\chi^2 = 23.024$, d.f. = 13, $p = 0.041$), with the lowest values being found in sites two and ten. Furthermore, mean pairwise *r* and sMLH were significantly negatively correlated (Pearson's correlation = −0.331, $p = 0.015$), indicating that sites with high levels of relatedness among genets also tended to have higher levels of inbreeding.

**Table 1.** Details of *B. edulis* sampling locations including the numbers of sporocarps and genets, the Shannon–Weiner diversity index, the Simpson's diversity index, expected heterozygosity (*He* or Nei's gene diversity) and the inbreeding coefficient (FIS) with significant values highlighted in italics. Also shown are mean Lynch and Ritland [31] relatedness values for all pairwise comparisons among genets within sample sites and the mean sMLH of the genets in each locality (±s.e.).

| sample site | site name | mean tree age in years (s.e.) | number of sporocarps | number of genets | Shannon's H | Simpson's index | *He* | FIS | mean relatedness | mean sMLH (s.e.) |
|---|---|---|---|---|---|---|---|---|---|---|
| 1 | Hallowed site | 76.07 (4.67) | 38 | 6 | 1.79 | 0.83 | 0.72 | 0.24 | 0.077 | 1.401 (0.442) |
| 2 | Gite | 23.33 (1.70) | 20 | 14 | 2.64 | 0.93 | 0.42 | −0.03 | 0.142 | 0.831 (0.395) |
| 3 | Leaning stick | 50.05 (4.65) | 18 | 3 | 1.10 | 0.67 | 0.47 | 0.27 | 0.165 | 0.970 (0.000) |
| 4 | Corner site | 72.63 (5.35) | 14 | 4 | 1.39 | 0.75 | 0.52 | 0.21 | 0.106 | 0.970 (0.457) |
| 5 | Corner below gite | 70.60 (8.20) | 8 | 1 | 0 | 0 | 0.57 | 0 | — | 1.293 (−) |
| 6 | Bauernhaus | 189.14 (11.26) | 7 | 1 | 0 | 0 | 0.43 | 0 | — | 0.970 (−) |
| 7 | Lone birch | 35.00 (−) | 6 | 4 | 1.39 | 0.75 | 0.42 | 0.40 | 0.175 | 0.808 (0.417) |
| 8 | Below Vincent's | 49.18 (5.66) | 5 | 5 | 1.61 | 0.80 | 0.43 | −0.18 | 0.139 | 1.164 (0.177) |
| 9 | Bauernhaus museum | 44.34 (4.13) | 4 | 4 | 1.39 | 0.75 | 0.43 | −0.91 | 0.273 | 1.131 (0.417) |
| 10 | Nico's place | 67.26 (5.30) | 4 | 4 | 1.39 | 0.75 | 0.31 | 0.50 | 0.140 | 0.323 (0.264) |
| 11 | Rosen | 47.57 (6.11) | 3 | 3 | 1.10 | 0.67 | 0.41 | 0.04 | 0.101 | 0.970 (0.000) |
| 12 | Scraggy site | 52.17 (7.13) | 4 | 3 | 1.10 | 0.67 | 0.66 | −0.20 | 0.045 | 1.509 (0.673) |
| 13 | Jacques | 183.69 (12.78) | 2 | 1 | 0 | 0 | 0.86 | 0 | — | 1.940 (−) |
| 14 | The dell | 50.53 (4.50) | 1 | 1 | 0 | 0 | 0.29 | 0 | — | 0.647 (−) |
| total | — | — | 134 | 54 | 3.94 | 0.98 | 0.62 | 0.23 | — | 1.000 |

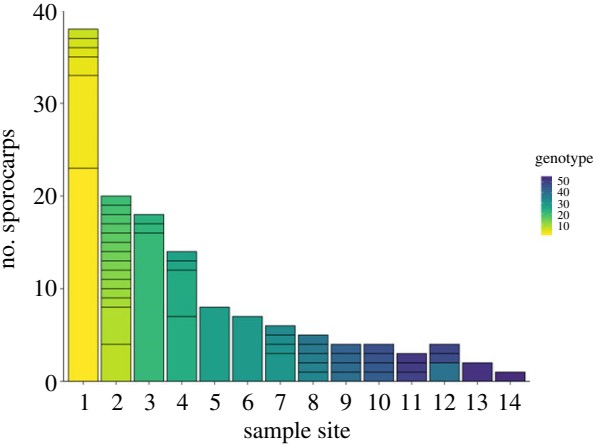

**Figure 3.** The distribution of *B. edulis* genet abundances among and within sample sites. Individual genets are colour coded along a continuum from the first to the 54th unique genotype.

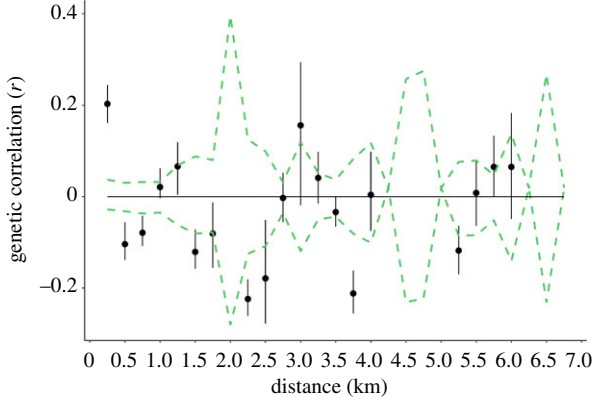

**Figure 4.** Spatial autocorrelation plot of the genetic correlation coefficient (*r*) as a function of geographic distance. Permuted 95% confidence intervals are depicted as dashed green lines together with bootstrapped 95% confidence error bars.

## 2.4. Relationship between genetic diversity and woodland age

To investigate the relationship between genetic diversity in *B. edulis* and woodland age, we fitted a series of generalized linear models. First, we expressed diversity as the number of genets relative to the total number of sporocarps collected from each sampling site and analysed this using a Poisson distribution to account for nonlinearity. Across the full dataset, the number of genets per site was significantly negatively associated with mean woodland age ($p = 0.009$, figure 6 and electronic supplementary material, table S4*a*). Including the main tree species present within each site as an additional three-level factor (1 = beech, 2 = birch and 3 = oak) resulted in a strengthening of the *p*-value associated with woodland age ($p = 0.001$, electronic supplementary material, table S5) and an increase in the proportion of explained variance from 53% to 68%. Similar relationships were found for two alternative diversity measures, Shannon's H and the Simpson index (electronic supplementary material, figure S2 and table S4*b* and *c*, respectively, *p*-values both <0.05). Mean pairwise relatedness was also significantly higher in younger sites (electronic supplementary material, figure S2 and table S4*d*, $p = 0.028$), and sMLH showed an inverse relationship with age, but this was not significant (electronic supplementary material, table S4*e*).

## 3. Discussion

Here we report a genetic analysis of *B. edulis* sporocarps sampled from multiple localities around Bielefeld in northwestern Germany. We uncovered important deviations from the null model of panmixia driven by high levels of wind dispersal. Specifically, at a local level, relatedness can be but

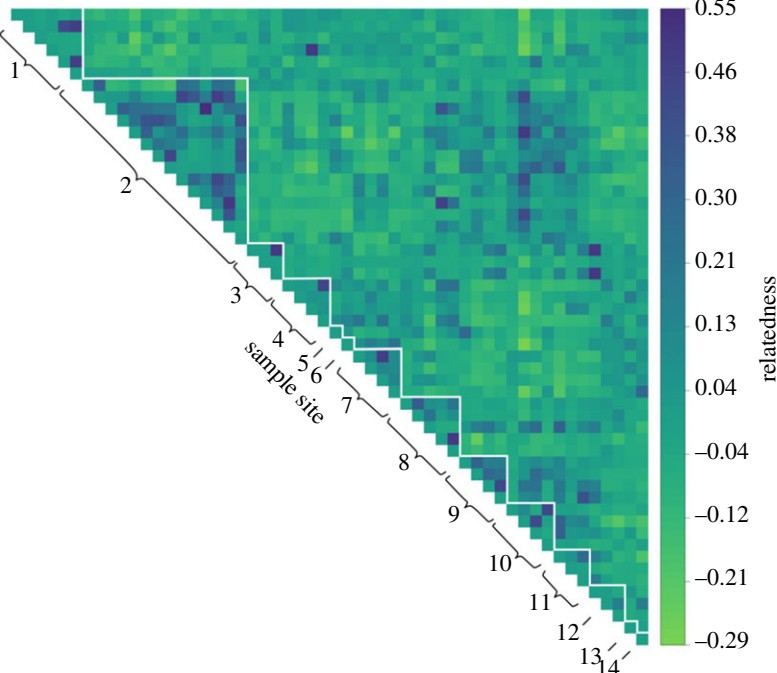

**Figure 5.** Heatmap of pairwise Lynch and Ritland relatedness (*r*) values among *B. edulis* genets. The individual genets are ordered from 1 to 54 along the *x* and *y* axes. The sampling sites are outlined in white and labelled from one to 14 along the diagonal.

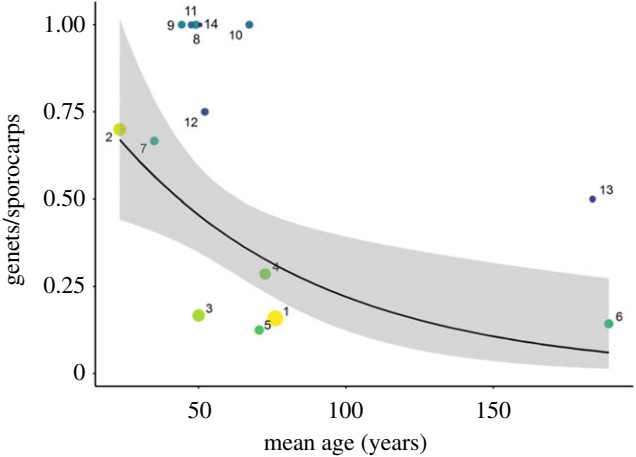

**Figure 6.** Relationship between number of genets (weighted by the number of sporocarps) at each sample site and mean tree age. The black trendline shows the fit of a generalized linear model with a Poisson distribution to the raw data, while the grey area represents the associated 95% confidence interval. Individual sample sites are represented as circles, which are colour coded according to the scheme shown in figure 3 and sized in proportion to the number of sporocarps.

is not always high, while genet diversity declines with the age of the trees in the stand. By implication, as new habitat becomes available, large numbers of genets may become established, but as the habitat matures, intra-specific competition reduces the chances of new spores adding to the population and may cause many genets to be lost. Our results open a new window on how *B. edulis* populations may develop over time.

## 3.1. Clonality

Each genet was on average represented by 2.5 sporocarps (range = 1.43 for the youngest site to 7 for the oldest site). This is comparable but perhaps a little more extreme than reported by Zhou *et al.* [9] who

studied *Suillus grevillei* and found ratios of 3.7 and 1.8 in larch stands that were 35 and 85 years old, respectively. Similarly, Lee and Koo [15] published ratios of between 1.2 and 2.3 sporocarps per genet in *Suillus granulatus*. These values are broadly similar to ours but direct comparisons are hampered both by observation bias and by the difficulty of controlling for changes in genetic diversity as hosts age, which may differ between species. Observation bias relates to the thoroughness of sampling. Porcini growing under beech can be cryptic and many sporocarps are variously picked by humans and eaten by a variety of animals. By contrast, *Suillus* species are much less picked by humans and, often growing in grass, can be more visible, making complete sampling more likely.

## 3.2. Population structure and genet diversity

We found little evidence of population structure among sites separated by a few kilometres, with only two pairwise comparisons yielding statistically significant $F_{st}$ values. In both cases, one of the populations was the youngest, most genetically diverse site (site two), which also had the second largest sample size of sporocarps, while the other two populations were those with the first and fourth largest sample sizes of sporocarps, respectively. This may suggest that the power to detect significant differences depends on the genetic diversity of the populations in question, and that overall power is quite low with only seven microsatellites. Our results are broadly in line with studies of other species such as *Tricholoma matsutake*, *Russula breviceps* and *Laccaria amethystina*, none of which showed evidence of genetic structure on a geographical scale of around 1 km [3,4,12]. However, if future studies confirm that the strongest structure is associated with younger stands, this would be surprising as it goes against the idea that these represent recent colonization events by wind-blown spores. The fact that older stands show no structure even after competition/selection has reduced diversity hints at the interesting possibility that similar genotypes may be independently successful in different locations. Alternatively, somatic incompatibility could potentially operate to reduce diversity by selecting against spores that are not genetically compatible with the genotypes of mycelia that are already established in the local area [16].

At odds with the broader scale pattern, spatial autocorrelation analysis uncovered significant fine-scale structuring but only over the smallest distance class (250 m). This contrasts with *Suillus spraguei* where no fine-scale clustering of similar genotypes was detected, indicating that there may be few restrictions to gene flow in this species [11]. However, locally elevated relatedness was observed in both *Cantherellus formosus* at distances up to 400 m [16] and in *Tricholoma scalpturatum* at distances up to 20 m [32]. These differing patterns across species suggest either that multiple mechanisms operate to shape local population structure or that each species' ecology has a strong influence on the genetic outcome.

High relatedness of sporocarps could arise either though the local recruitment of spores to the same population or if individual mycelia reproduce by interacting with multiple others. These possibilities could be distinguished either by investigating the spatial distribution of pairs of close relatives, with distances above 30–50 m being more likely to reflect wind or insect-dispersed spores, or by sampling the mycelia themselves so as to identify the underlying haploid genotypes. Having said this, a general failure to find evidence of strong population structure over larger scales might argue against the idea that most spores recruit successfully on a scale of tens to hundreds of metres, as this would tend to generate strong local genetic structure that would, over time, extend up to larger spatial scales. For *B. edulis*, therefore, our data suggest that while structure can develop over short temporal and spatial scales, the longer term outcome appears to be influenced by factors that act to prevent these patterns extending, such as mycelial competition, selection or even occasional years when long-distance spore dispersal rates are high.

## 3.3. Woodland age effects

Previous studies of *Suillus bovinus*, *S. grevillei* and *Russula brevipes* have shown by comparing stands of trees of different age that fungal clone size increases and the number of clones decreases with increasing forest age [8–10]. This is the same pattern that we find, but we have many more sites in total (14 versus 2–4), allowing us to begin to consider the process more as a continuum. Understanding this process appears critical to understanding how individual populations develop as well as how the species as a whole evolves. However, our study system is currently somewhat limited, both because we have already sampled the majority of locally available sites and because the sites themselves are managed, including some degree of thinning of the trees. In addition, we did not record the exact locations of

the sporocarps within sites, precluding testing of the hypothesis that older woodland patches should contain larger genets. Nonetheless, our findings are important in that they point towards future sampling regimes that may allow the process to be unpicked. In particular, it would be useful to develop methods to sample mycelia and thereby directly measure the sizes of individual fungal mats, as well as to find older stands of trees that will help define the climax state.

In order to better understand the mechanisms driving the association between genet diversity and woodland age, it will also be important to consider genet longevity. For instance, in older forests, when one mycelium dies, does this reset the clock and allow the process to begin again with high diversity recruitment, or do younger mycelia expand to occupy the available space? One potential factor influencing the longevity of individual genets could be heterozygosity, as Gryta *et al*. [33] have shown that the inbred progeny of *Hebeloma cylindrosporum* are often short-lived relative to outcrossed mycelia. This generates the prediction that heterozygosity should increase as genet diversity decreases. Our data are too limited to test for such a pattern meaningfully, because many of our sites yielded only a single genotype. However, if future research vindicates this prediction, it would provide a simple explanation for why population substructure does not develop: selection for high heterozygosity favours the descendants of long-distance dispersal events so will always act against processes that tend to generate local structure. To quantify inbreeding more precisely will require the development of larger panels of genetic markers from across the genome [34].

In addition, fungi often experience genomic changes during vegetative growth, including mitotic recombination and loss of heterozygosity [35]. It is unclear how common these processes are in *B. edulis*, but they have the potential to cause a reduction in individual heterozygosity as a stand ages. However, the effect does not seem to be large in our sample because, ironically, loss of heterozygosity will tend to increase genotype diversity, the exact opposite trend to the one we observe. Specifically, an event that causes a locus to become homozygous will effectively split that genet into two, one which is homozygous at the effected locus and the other of which is heterozygous. In the future, we aim to analyse much larger sample sizes and to extend our panel of genetic markers, both of which should allow us to document these events should they be present at any appreciable frequency.

## 3.4. Strengths and weaknesses of our study

Compared with studies of other species, our study benefits from a relatively large number of different, discrete sites and their accessibility, which allows more or less exhaustive sampling throughout the season. The use of highly variable microsatellites also allowed us to resolve individual genets, even when relatedness was high, which is an advantage over the use of less polymorphic markers such as internal transcribed spacers. Indeed, while it would be desirable to increase the number of markers further, we were unable to find previous studies where pairwise relatedness between genets could be or had been quantified. This possibility is likely to prove important in understanding fine-scale patterns of reproduction and resolving the balance between sexual and vegetative reproduction. Increasing the number of genetic markers should also allow us to ask questions about inbreeding and the importance of outcrossing.

Our study also has a number of limitations. In particular, we assume that sporocarp diversity provides a reasonable proxy for mycelial diversity. Consequently, we cannot distinguish between an absence of fruit indicating that a particular genet has been outcompeted and died versus an absence of fruit indicating that a genet is focusing its resources on maximizing vegetative growth rather than producing sporocarps. In our experience, in 'good years', sporocarps appear in a greater diversity of sites compared with 'poor years', with fruit appearing near certain trees only in exceptional years. Consequently, short-term changes in sporocarp diversity may, in reality, merely reflect differing propensities for fruiting. If so, by sampling only in a single season, we may have missed the opportunity to detect genets that only fruit in unusually good years. It will be interesting in future studies to see whether inter-annual patterns help to resolve questions about how intra-specific competition operates. For example, do good years reveal a higher diversity of genets in older stands of trees, thereby showing that mycelia persist and tend not to fruit rather than dying?

## 3.5. Conclusion

Our study of *B. edulis* reveals a rather dynamic picture in which new stands of trees are colonized by high diversity genets, whose representation progressively diminishes with increasing tree age. As yet, it is unclear whether this pattern reflects actual mycelial mortality or simply a failure of certain genets to

garner enough resources in competition with the dominant genets. To understand this process better will require the sampling of mycelia rather than just fruit, coupled with genome-wide sequence data to reveal if and where selection is acting.

# 4. Materials and methods

## 4.1. Sample collection

A total of 134 *B. edulis* sporocarps (table 1) were sampled during 2015 from around Bielefeld in northwest Germany. Bielefeld is approximately 118 m above sea level, the temperature averages 8.9°C and the average annual rainfall is 832 mm. The study area (figure 2) lies mainly to the west of the city and comprises multiple patches of mixed deciduous woodland. The most abundant tree species are beech (*Fagus sylvatica*) followed by common oak (*Quercus robur*) and silver birch (*Betula pendula*). The sample sites (figure 2 and table 1) are mainly dominated by beech, with the exception of sites six and 13, which are pure oak stands, and site seven which contains a solitary silver birch. Because the majority of sample sites were small (approximately 50–100 m$^2$) in relation to the accuracy of civilian GPS (approximately 5–10 m), we recorded a single GPS location for each site using a Garmin 64 s handheld navigator. As a proxy for the age of the woodland in each sample site, we measured the circumference of each tree greater than 20 cm in diameter and also recorded the species of each tree. The resulting data were then fed into a tree age calculator (http://www.tree-guide.com/tree-age-calculator) to ascertain the approximate age of each tree (to within ±10%) and from there to derive a mean ± s.e. for each site. In order to sample the sporocarps as comprehensively as possible, all of the sites were visited between two and four times a week from mid-August until mid-October 2015. The first and last sporocarps were found on 29 August and 10 October 2015, respectively.

## 4.2. Development and testing of *B. edulis* microsatellites

We downloaded the *B. edulis* genome sequence version 1.0 from https://mycocosm.jgi.doe.gov/Boled1/Boled1.home.html and searched for perfect di- tri- and tetranucleotide motifs using MISA-web [36]. Primers were designed for microsatellites with unusually large numbers of repeats (*B. edulis* tends to have short microsatellites, so in practice this meant more than eight pure repeats for dinucleotides, and more than four pure repeats for tri- and tetranucleotides) and where the flanking regions were not themselves overly repetitive, as judged by eye. Primers were designed using Primer3Plus [37] with default settings except for the specification of an optimal primer melting temperature of 60°C. Testing was then conducted in multiplexes on a panel of 96 *B. edulis* samples, and primers revealing three or more alleles were combined into multiplexes for genotyping.

## 4.3. Microsatellite genotyping

Total genomic DNA was extracted using a standard Proteinase K digestion (in: 1% SDS, 50 mM TrisHC Ph 8.0, 5 mM EDTA) followed by purification with glass milk. Briefly, 50 µl of digestate was added to 100 µl of 6 M NaI and 5 µl glass milk. After 20–30 min adsorption time, the glass milk was spun down and washed twice with 'new wash solution' (100 mM NaCl, 1 mM EDTA, 10 mM Tris-HCl pH 8, 50% EtOH) before the DNA was eluted in 150 µl low TE buffer. For speed and efficiency, all operations were carried out in 96-well microtitre plates. For each PCR reaction, 1 µl of DNA was added to 3 µl H$_2$O and 4 µl Qiagen 2X multiplex PCR mastermix to which PCR primers had already been added. All of the primers were multiplexed into two separate multiplexed reactions using the following PCR profile: one cycle of 5 min at 95°C; 36 cycles of 45 s at 95°C, 3 min at 58°C and 1 min at 68°C; and one final cycle of 10 min at 72°C. The resulting PCR products were then resolved by electrophoresis on an ABI 3730xl and allele sizes were called relative to the LIZ size standard (1 : 150 dilution) using GeneMarker version 2.6.2 (SoftGenetics®: State College, Pennsylvania, USA). Samples with missing genotypes at three or more loci were excluded from further analysis.

## 4.4. Microsatellite summary statistics

All of the loci were tested for deviation from Hardy–Weinberg equilibrium (HWE) using Genepop v. 4.2 [38]. For each test, we set the dememorization number to 10 000, the number of batches to 1000 and the

number of iterations per batch to 10 000. False discovery rate (FDR) corrections [39] with an $\alpha$ level of 0.05 were applied to the resulting $p$-values to account for multiple testing. Genepop was also used to calculate observed and expected heterozygosities as well as to determine the number of alleles amplified at each locus.

## 4.5. Identification of unique genets

To control for clonal replicates, all identical multilocus genotypes were removed from the full dataset using the clonecorrect function in the R package poppr v. 2.8.3 [40,41]. To provide a measure of the power of the genetic markers to discriminate individual clones, the probability of identity was calculated across all loci using GenAlEx v. 6.5 [42,43].

## 4.6. Genetic diversity and differentiation

The Shannon–Weiner diversity index, the Simpson's diversity index and expected heterozygosity ($He$, Nei's gene diversity) were calculated using the poppr function in the R package poppr. F-statistics among mushrooms from the 14 sites were computed using Arlequin v. 3.5.2.2 [44,45] with 10 000 permutations. The missing data threshold was set at 5%, limiting the number of usable loci for distance computation to three (AAC92, AC8 and AC111). The resulting significance values were FDR corrected as described above.

## 4.7. Spatial autocorrelation analysis

Autocorrelation analysis was conducted using the genotypes of 54 unique *B. edulis* genets. We used spatial autocorrelation techniques developed by Smouse and colleagues [46–48] which employ a multivariate approach to assess the spatial signal captured by multiple codominant microsatellite loci. This analysis was performed using GenAlEx. This software calculates an autocorrelation coefficient $r$ using two pairwise matrices, one containing geographic distances and the other containing squared genetic distances [47]. The autocorrelation coefficient is calculated for a specified number of distance classes and provides a measure of the genetic similarity between pairs of individuals falling within each distance class. We selected a distance class of 250 m to allow the examination of fine-scale patterns of genetic relatedness across our study area. Spatial genetic autocorrelograms were produced by plotting $r$ as a function of geographic distance. Tests for statistical significance were performed as described by Peakall *et al.* [48]. Two methods were used: random permutation and bootstrap estimates of $r$, with the number of permutations and bootstraps set to 1000. For small samples, bootstrap errors tend to be larger than permutational errors, and consequently bootstrap tests are more conservative and will favour the null hypothesis more frequently than permutational tests. Here, we reported both the results of bootstrap and permutational tests, but declared significance only when both tests indicated a significant result at $p < 0.05$.

## 4.8. Quantification of pairwise relatedness values and inbreeding

Pairwise relatedness among the 54 *B. edulis* genets was estimated according to Lynch and Ritland [31], for which the relatedness coefficient between two individuals is defined in terms of probabilities of identity by descent. Given that relatedness coefficients depend on a reference, negative values indicate that individuals are less related to each other than on average. This analysis was performed using GenAlEx and the results were visualized using the R package corrplot version 0.84 [49]. To investigate inbreeding, we then used the R package inbreedR [50] to quantify the magnitude of identity disequilibrium (ID) by calculating the two-locus heterozygosity disequilibrium, $g_2$, [51] and its 95% confidence interval through 1000 permutations of the microsatellite dataset. Heterozygosity was then quantified for each genet using the measure standardized multilocus heterozygosity (sMLH), which quantifies the proportion of loci that are heterozygous while weighting the contribution of each locus by the expected heterozygosity at that locus [52].

## 4.9. Relationship between genetic diversity and woodland age

To investigate the relationship between the number of *B. edulis* genets within sample sites and mean tree age, a generalized linear model (GLM) was constructed with a Poisson distribution and with

weights equal to the number of sporocarps at each site. To investigate the relationship between mean tree age and Shannon's H, Simpson's index, mean sMLH and mean pairwise relatedness, we constructed four independent GLMs with Gaussian distributions. All models were built in base R v. 3.5.3 and visualized with the attached packages car version 3.0.3 [53], ggplot2 v. 3.2.1 [54] and sjPlot v. 2.7.2 [55].

Data accessibility. The clone-corrected microsatellite dataset is available via the Dryad Digital Repository (https://dx.doi.org/10.5061/dryad.1g1jwstrw) [56]. The code used to analyse the data and accompanying documentation are available as a PDF file written in Rmarkdown (electronic supplementary material, File S1).

Authors' contributions. J.I.H. and W.A. conceived the study. J.I.H. collected the samples. D.A.W. conducted the laboratory work and scored the genotypes. R.N., V.L. and J.I.H. analysed the data. J.I.H. and W.A. wrote the manuscript and all of the authors edited and approved the final manuscript.

Competing interests. We declare we have no competing interests.

Funding. R.N. was funded by the Deutsche Forschungsgemeinschaft (DFG, German Research Foundation) in the framework of a Sonderforschungsbereich (project nos. 316099922 and 396774617–TRR 212). D.A.W. was funded by DFG standard grant (HO 5122/5-1). The article processing charge was funded by the DFG and the Open Access Publication Fund of Bielefeld University.

Acknowledgements. We are grateful to Bryn Dentinger and two anonymous referees for helpful comments on our manuscript.

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
