## [Reviewer comments · Royal Society Open Science]

Review History

RSOS-200419.R0 (Original submission)

Review form: Reviewer 1

Is the manuscript scientifically sound in its present form?

No

Are the interpretations and conclusions justified by the results?

Yes

Is the language acceptable?

Yes

Do you have any ethical concerns with this paper?

No

Have you any concerns about statistical analyses in this paper?

No

Recommendation?

Accept with minor revision (please list in comments)

Comments to the Author(s)

The manuscript titled "Genetic analysis of *Boletus edulis* suggests that intra-specific competition may reduce local genetic diversity as a woodland ages" faces the genetic diversity of the worldwide appreciated fruiting bodies of *Boletus edulis* through microsatellites searched in its genome. The work is interesting and detailed for what concerns the analysis using the R package, and less detailed towards the biology and taxonomy of the *Boletus*. I would like to see in the Introduction some word on the taxonomy of this species. How the Authors have identified the fruitbodies? Just from their morphological features? I think that a molecular identification by sequencing the ITS region is necessary. For both purposes, I suggest Mello et al. 2006. Have the analysed samples been deposited in Herbarium? This is a common practice in mycology to allow reproducible science. Another crucial point is why investigate the intraspecific variability of *Boletus edulis*. It's true that this fungus is collected wild but I suggest to add in the Introduction that many attempts to cultivate this fungus are ongoing (Mediavilla et al. 2016) and that understanding its population structure would be useful in programmes aimed to reach its domestication.

Did the Authors know that there were microsatellites already available (Wang et al 2015)?

The hypothesis that competition between genets may reduce genetic diversity is well argued by the authors but needs to be tested to its mycelium as the authors themselves state in the conclusion.

Review form: Reviewer 2 (JP Xu)

Is the manuscript scientifically sound in its present form?

Yes

Are the interpretations and conclusions justified by the results?

Yes

Is the language acceptable?

Yes

Do you have any ethical concerns with this paper?

No

Have you any concerns about statistical analyses in this paper?

No

Recommendation?

Accept with minor revision (please list in comments)

Comments to the Author(s)

This is a well-written paper, easy to read. The data analyses were adequate and the conclusions, though tentative, seemed overall justified. However, I do have a couple of concerns and suggestions.

1. Sites 6 and 13 differ significantly from others in their dominant tree species. These two sites also have the most different genotype diversity from others with most correlational statistics driven by these two sites. Is it possible that the observed correlation was not due to age of forest but forest tree species?

2. I would like to see estimates of genet sizes for each of the sites, at least for the largest genet at each site, based on the one year sample. If age is a factor, we should see older forests having larger genets.
3. In fungi, aside from inbreeding, mitotic recombination during asexual propagation could also cause the reduction in heterozygosity. In older forests, the possibility of mitotic recombination may be higher.

Figure 3: As the total is >100%, the Y-axis should be sporocarp counts/genet.

Decision letter (RSOS-200419.R0)

Dear Ms Litzke

On behalf of the Editors, I am pleased to inform you that your Manuscript RSOS-200419 entitled "Genetic analysis of *Boletus edulis* suggests that intra-specific competition may reduce local genetic diversity as a woodland ages" has been accepted for publication in Royal Society Open Science subject to minor revision in accordance with the referee suggestions. Please find the referees' comments at the end of this email.

The reviewers and handling editors have recommended publication, but also suggest some minor revisions to your manuscript. Therefore, I invite you to respond to the comments and revise your manuscript.

- Ethics statement

- Data accessibility

<http://datadryad.org/submit?journalID=RSOS&manu=RSOS-200419>

- Competing interests

- Authors' contributions

- Acknowledgements

- Funding statement

Because the schedule for publication is very tight, it is a condition of publication that you submit the revised version of your manuscript before 27-May-2020. Please note that the revision deadline will expire at 00.00am on this date. If you do not think you will be able to meet this date please let me know immediately.

- 1) A text file of the manuscript (tex, txt, rtf, docx or doc), references, tables (including captions) and figure captions. Do not upload a PDF as your "Main Document";

- 2) A separate electronic file of each figure (EPS or print-quality PDF preferred (either format should be produced directly from original creation package), or original software format);
- 3) Included a 100 word media summary of your paper when requested at submission. Please ensure you have entered correct contact details (email, institution and telephone) in your user account;
- 4) Included the raw data to support the claims made in your paper. You can either include your data as electronic supplementary material or upload to a repository and include the relevant doi within your manuscript. Make sure it is clear in your data accessibility statement how the data can be accessed;
- 5) All supplementary materials accompanying an accepted article will be treated as in their final form. Note that the Royal Society will neither edit nor typeset supplementary material and it will be hosted as provided. Please ensure that the supplementary material includes the paper details where possible (authors, article title, journal name).

If your manuscript is newly submitted and subsequently accepted for publication, you will be asked to pay the article processing charge, unless you request a waiver and this is approved by Royal Society Publishing. You can find out more about the charges at <https://royalsocietypublishing.org/rsos/charges>. Should you have any queries, please contact openscience@royalsociety.org.

Kind regards,

Anita Kristiansen
Editorial Coordinator

on behalf of Dr Punidan Jeyasingh (Associate Editor) and Kevin Padian (Subject Editor)
openscience@royalsociety.org

Associate Editor Comments to Author (Dr Punidan Jeyasingh):

Comments to the Author:

This paper was reviewed by two experts, both of whom were highly enthusiastic about it. They make a few minor comments that the authors should address. I felt the reviews were constructive. I invite the authors to submit a fresh version incorporating these comments.

Reviewer comments to Author:

Reviewer: 1

Comments to the Author(s)

The manuscript titled "Genetic analysis of *Boletus edulis* suggests that intra-specific competition may reduce local genetic diversity as a woodland ages" faces the genetic diversity of the worldwide appreciated fruiting bodies of *Boletus edulis* through microsatellites searched in its genome. The work is interesting and detailed for what concerns the analysis using the R package, and less detailed towards the biology and taxonomy of the *Boletus*. I would like to see in the Introduction some word on the taxonomy of this species. How the Authors have identified the fruitbodies? Just from their morphological features? I think that a molecular identification by sequencing the ITS region is necessary. For both purposes, I suggest Mello et al. 2006. Have the analysed samples been deposited in Herbarium? This is a common practice in mycology to allow reproducible science. Another crucial point is why investigate the intraspecific variability of *Boletus edulis*. It's true that this fungus is collected wild but I suggest to add in the Introduction that many attempts to cultivate this fungus are ongoing (Mediavilla et al. 2016) and that understanding its population structure would be useful in programmes aimed to reach its domestication.

Did the Authors know that there were microsatellites already available (Wang et al 2015)?

The hypothesis that competition between genets may reduce genetic diversity is well argued by the authors but needs to be tested to its mycelium as the authors themselves state in the conclusion.

Reviewer: 2

Comments to the Author(s)

This is a well-written paper, easy to read. The data analyses were adequate and the conclusions, though tentative, seemed overall justified. However, I do have a couple of concerns and suggestions.

1. Sites 6 and 13 differ significantly from others in their dominant tree species. These two sites also have the most different genotype diversity from others with most correlational statistics driven by these two sites. Is it possible that the observed correlation was not due to age of forest but forest tree species?
2. I would like to see estimates of genet sizes for each of the sites, at least for the largest genet at each site, based on the one year sample. If age is a factor, we should see older forests having larger genets.
3. In fungi, aside from inbreeding, mitotic recombination during asexual propagation could also cause the reduction in heterozygosity. In older forests, the possibility of mitotic recombination may be higher.

Figure 3: As the total is >100%, the Y-axis should be sporocarp counts/genet.

Author's Response to Decision Letter for (RSOS-200419.R0)

See Appendix A.

Decision letter (RSOS-200419.R1)

Dear Ms Litzke:

On behalf of the Editors, I am pleased to inform you that your Manuscript RSOS-200419.R1 entitled "Genetic analysis of *Boletus edulis* suggests that intra-specific competition may reduce local genetic diversity as a woodland ages" has been accepted for publication in Royal Society Open Science subject to minor revision in accordance with the referee suggestions. Please find the referees' comments at the end of this email.

The reviewers and Subject Editor have recommended publication, but also suggest some minor revisions to your manuscript. Therefore, I invite you to respond to the comments and revise your manuscript.

- Ethics statement

- Data accessibility

<http://datadryad.org/submit?journalID=RSOS&manu=RSOS-200419.R1>

- Competing interests

- Authors' contributions

- Acknowledgements

- Funding statement

Because the schedule for publication is very tight, it is a condition of publication that you submit the revised version of your manuscript before 19-Jun-2020. Please note that the revision deadline will expire at 00.00am on this date. If you do not think you will be able to meet this date please let me know immediately.

Supplementary files will be published alongside the paper on the journal website and posted on the online figshare repository (<https://figshare.com>). The heading and legend provided for each supplementary file during the submission process will be used to create the figshare page, so please ensure these are accurate and informative so that your files can be found in searches. Files

on figshare will be made available approximately one week before the accompanying article so that the supplementary material can be attributed a unique DOI.

Kind regards,

Anita Kristiansen
Editorial Coordinator

on behalf of Dr Punidan Jeyasingh (Associate Editor) and Kevin Padian (Subject Editor)
openscience@royalsociety.org

Associate Editor Comments to Author (Dr Punidan Jeyasingh):

Comments to the Author:

I thank the authors for carefully addressing reviewer comments. There is only one outstanding issue. The Wang et al. (2015) reference. I am asking Reviewer 1 for the proper reference so that the authors can track that paper down and, if suitable, incorporate relevant information (e.g., microsat markers).

Author's Response to Decision Letter for (RSOS-200419.R1)

See Appendix B.

Decision letter (RSOS-200419.R2)

Dear Ms Litzke,

It is a pleasure to accept your manuscript entitled "Genetic analysis of *Boletus edulis* suggests that intra-specific competition may reduce local genetic diversity as a woodland ages" in its current form for publication in Royal Society Open Science.

Kind regards,
Lianne Parkhouse
Editorial Coordinator
Royal Society Open Science
openscience@royalsociety.org

on behalf of Dr Punidan Jeyasingh (Associate Editor) and Kevin Padian (Subject Editor)
openscience@royalsociety.org

Appendix A

Universität Bielefeld

Faculty for Biology
Department of Animal Behaviour
Joseph Hoffman PhD

University of Bielefeld | Postfach 10 01 31 | 33501 Bielefeld

Morgenbreede 45
33615 Bielefeld
Room VHF 203
Telephone: (0521) 106 - 2711
Telefax: (0521) 106 - 2998
e-mail: joseph.hoffman@uni-bielefeld.de
18th May 2020

Editor-in-Chief, *Royal Society Open Science*

Dear Prof Sanders,

Please find attached our revised manuscript '**Genetic analysis of *Boletus edulis* suggests that intra-specific competition may reduce local genetic diversity as a woodland ages**'. A short response to the referee's comments is provided below.

Please do not hesitate to contact us if you require anything further.

Yours sincerely,

Prof. Dr. Joseph Hoffman and colleagues

Dear Ms Litzke

On behalf of the Editors, I am pleased to inform you that your Manuscript RSOS-200419 entitled "Genetic analysis of *Boletus edulis* suggests that intra-specific competition may reduce local genetic diversity as a woodland ages" has been accepted for publication in Royal Society Open Science subject to minor revision in accordance with the referee suggestions. Please find the referees' comments at the end of this email.

The reviewers and handling editors have recommended publication, but also suggest some minor revisions to your manuscript. Therefore, I invite you to respond to the comments and revise your manuscript.

Ethics statement

Response 1. An ethical statement does not apply as the study was conducted on a common edible fungus and not on animals or humans.

Data accessibility

<http://datadryad.org/submit?journalID=RSOS&manu=RSOS-200419>

Response 2. We have uploaded the microsatellite data to Dryad and have updated the data accessibility section accordingly.

Competing interests

Response 3. This is present in the correct format.

Authors' contributions

All submissions, other than those with a single author, must include an Authors' Contributions section which individually lists the specific contribution of each author. The list of Authors should meet all of the following criteria; 1) substantial contributions to conception and design, or acquisition of data, or analysis and interpretation of data; 2) drafting the article or revising it critically for important intellectual content; and 3) final approval of the version to be published. All contributors who do not meet all of these criteria should be included in the acknowledgements. We suggest the following format:

Response 4. This is present in the correct format.

Acknowledgements

Response 5. We have moved funding information to a new funding statement.

- Funding statement

Response 6. We have done this.

Response 7. All of these sections have been provided at the end of the manuscript in the order that is shown in the screenshot.

Because the schedule for publication is very tight, it is a condition of publication that you submit the revised version of your manuscript before 27-May-2020. Please note that the revision deadline will expire at 00.00am on this date. If you do not think you will be able to meet this date please let me know immediately.

Supplementary files will be published alongside the paper on the journal website and posted on the online figshare repository (<https://rs.figshare.com/>). The heading and legend provided for each supplementary file during the submission process will be used to create the figshare page, so please ensure these are accurate

and informative so that your files can be found in searches. Files on figshare will be made available approximately one week before the accompanying article so that the supplementary material can be attributed a unique DOI.

If your manuscript is newly submitted and subsequently accepted for publication, you will be asked to pay the article processing charge, unless you request a waiver and this is approved by Royal Society Publishing. You can find out more about the charges at <https://royalsocietypublishing.org/rsos/charges>. Should you have any queries, please contact openscience@royalsociety.org.

Kind regards,

Anita Kristiansen
Editorial Coordinator

on behalf of Dr Punidan Jeyasingh (Associate Editor) and Kevin Padian (Subject Editor)
openscience@royalsociety.org

Associate Editor Comments to Author (Dr Punidan Jeyasingh):

Comments to the Author:

This paper was reviewed by two experts, both of whom were highly enthusiastic about it. They make a few minor comments that the authors should address. I felt the reviews were constructive. I invite the authors to submit a fresh version incorporating these comments.

Reviewer comments to Author:

Reviewer: 1

Comments to the Author(s)

The manuscript titled “Genetic analysis of *Boletus edulis* suggests that intra-specific competition may reduce local genetic diversity as a woodland ages“ faces the genetic diversity of the worldwide appreciated fruiting bodies of *Boletus edulis* through microsatellites searched in its genome. The work is interesting and detailed for what concerns the analysis using the R package, and less detailed towards the biology and taxonomy of the *Boletus*. I would like to see in the Introduction some word on the taxonomy of this species.

Response 8. We have included this together with appropriate references.

How the Authors have identified the fruitbodies? Just from their morphological features? I think that a molecular identification by sequencing the ITS region is necessary. For both purposes, I suggest Mello et al. 2006.

Response 9. The specimens were collected by people with collectively over two decades of experience of field identification. *B. edulis* is one of the easiest species to identify, which is why it is so widely picked. The only possible confusions are with:

a) *B. pinophilus*, which grows under pine and with which we are very familiar. We are collecting a parallel dataset on this species and, although the microsatellites mostly cross-amplify, several markers have entirely distinct alleles.

b) *B. reticulatus*, which generally occurs earlier, is much rarer, and is identifiable by its paler form and brownish rather than white netting on the stem.

c) *B. aereus* is again very rare, has a darker cap and reddish stem. We have tried genotyping dried mushrooms from Greece and these proved to be mainly *B. aereus*. From these samples, we were able to confirm that, like *B. pinophilus*, *B. aereus* has distinct microsatellite alleles that make confusion impossible.

Have the analysed samples been deposited in Herbarium? This is a common practice in mycology to allow reproducible science.

Response 10. We agree that this is a good idea, but it has not been done and cannot be done retrospectively. This project was started by geneticists who were interested in fungi but lacked familiarity with mycological customs. In fact, we are considering depositing some samples in herbaria in the future. However, there are several practicalities that need overcoming:

a) In our experience, herbaria prefer entire fruit bodies. When collecting, we are keen to sample as many individuals as possible but not to operate a 'scorched earth' policy of removing everything we find. Currently, we collect most samples by taking a thin slice out of the cap. This is a relatively non-invasive approach that allows the mushroom to carry on producing spores.

b) The initial work has led to the development of a much larger project which is being set up aimed at collecting thousands of samples. It is unclear that we have the drying resources or that herbaria have the space to deal with such large numbers. In our limited experience, herbaria are keener on receiving modest numbers of well-defined 'type specimens'.

Another crucial point is why investigate the intraspecific variability of *Boletus edulis*. It's true that this fungus is collected wild but I suggest to add in the Introduction that many attempts to cultivate this fungus are ongoing (Mediavilla et al. 2016) and that understanding its population structure would be useful in programmes aimed to reach its domestication.

Response 11. We have incorporated this point into the introduction. However, we could only find a project on research gate when we searched for 'Mediavilla et al 2016' and 'Boletus'. Because we could not find any trace of this paper, we have cited another.

Did the Authors know that there were microsatellites already available (Wang et al 2015)?

Response 12. Thank you for pointing this out. However, despite extensive online searches in Google, Google Scholar and on the Web of Science, we have been unable to find any signs of a publication by Wang et al (2015) containing information on *Boletus* microsatellites. The closest we came was a paper by Wang (2015) on 'Variations in Element Levels Accumulated in Different Parts of *Boletus edulis* Collected from Central Yunnan Province, China' but this did not contain any information on genetic markers.

The hypothesis that competition between genets may reduce genetic diversity is well argued by the authors but needs to be tested to its mycelium as the authors themselves state in the conclusion.

Response 13. Absolutely, and we are currently seeking funding to look into this and related questions.

Reviewer: 2

Comments to the Author(s)

This is a well-written paper, easy to read. The data analyses were adequate and the conclusions, though tentative, seemed overall justified.

Response 14. Thank you for the overall positive appraisal.

However, I do have a couple of concerns and suggestions.

1. Sites 6 and 13 differ significantly from others in their dominant tree species. These two sites also have the most different genotype diversity from others with most correlational statistics driven by these two sites. Is it possible that the observed correlation was not due to age of forest but forest tree species?

Response 15. This is a good point. To investigate, we took two approaches. First, we re-ran the GLM of genetic diversity after excluding sites 6 and 13. The p-value for age strengthened from $p = 0.009$ to $p = 0.001$. We then re-ran the GLM, this time including the main tree species present at each site as an addition three-level factor (1 = beech, 2 = birch and 3 = oak). This again resulted in a strengthening of the p -value associated with woodland age ($p = 0.001$, Table 1, below) and an increase in the total proportion of explained variance (r^2) from 53% to 68%. As it is preferable to include all of the data, we have added the second of these models to our results section together with an additional supplementary table detailing the results.

Table 2: Results of a generalized linear model of the number of *B. edulis* genets (weighted by the number of sporocarps) including the main tree species present within each site as a three-level factor. A Poisson distribution was used to account for non-linearity in the data.

(A)	Genets / sporocarps		
Predictors	Incidence Rate Ratios	CI	p
(Intercept)	1.41	0.66–2.80	0.344
Mean tree age	0.98	0.96–0.99	0.001
Main tree species birch	1.07	0.32–2.69	0.898
Main tree species oak	12.78	1.05–128.58	0.034
Observations	14		
R ² Nagelkerke	0.676		

2. I would like to see estimates of genet sizes for each of the sites, at least for the largest genet at each site, based on the one year sample. If age is a factor, we should see older forests having larger genets.

Response 16. Unfortunately, we did not record the locations of individual sporocarps within sampling sites (mainly because the sampling sites were small relative to the accuracy of a civilian GPS), so we are unable to estimate genet sizes. We have acknowledged this limitation in our revised discussion, although this does not alter our conclusions.

3. In fungi, aside from inbreeding, mitotic recombination during asexual propagation could also cause the reduction in heterozygosity. In older forests, the possibility of mitotic recombination may be higher.

Response 17. Thanks for raising this interesting point. However, this effect does not seem to be large in our sample because, ironically, loss of heterozygosity will tend to *increase* genotype diversity, the exact opposite trend to the one we observe. Specifically, any event that causes a locus to become homozygous will effectively split that genet into two, one which is homozygous at the effected locus and the other of which is heterozygous. We have incorporated this point into the discussion.

Figure 3: As the total is >100%, the Y-axis should be sporocarp counts/genet.

Response 18. Good point. We have changed the y axis to 'Number of sporocarps', which we think is a more accurate description of what is being depicted.

Appendix B
Vivienne Litzke <vivienne.litzke@gmail.com>

Royal Society Open Science - Decision on Manuscript ID RSOS-200419.R1

Joe Hoffman <j_i_hoffman@hotmail.com>

Mon, Jun 15, 2020 at 9:38 AM

To: Vivienne Litzke <vivienne.litzke@gmail.com>, Rebecca Nagel <rebecca.nagel@uni-bielefeld.de>, "openscience@royalsociety.org" <openscience@royalsociety.org>

Dear Royal Society Open Science editorial team,

I am writing to enquire about our recently accepted paper. We received the following communication (below) early last week. We are concerned that the publication of our manuscript may be delayed owing to the editor wishing to communicate with one of the reviewers about a comment they raised about a previous study that apparently also developed microsatellites for our study species, *B. edulis*. Two points are relevant in this regard:

First, we did in fact search for this paper but were unable to find it, suggesting to us that perhaps it was not formally published, or otherwise it might have come out in a thesis or in the grey literature. Second, even if someone else did develop microsatellites for *B. edulis*, we cannot see a clear argument for why we should cite their work, because our paper uses microsatellites that we developed ourselves. We work on long-term studies of other species, such as fur seals, in which numerous teams have developed >100 microsatellites. However, it would not make sense to cite every paper that developed a microsatellite for this species - instead, the usual practice is to cite only studies that developed the microsatellites that were specifically used. In essence, we cannot see any legitimate reason for citing marker development efforts that are not relevant to the current study because those markers were not used, unless these studies entail some form of conceptual contribution that is relevant to the manuscript in question.

We hope you can understand that we are keen to push onto the proofs stage as quickly as possible and to avoid unnecessary delays.

With best wishes,

Joe Hoffman and colleagues

Comments to the Author:

I thank the authors for carefully addressing reviewer comments. There is only one outstanding issue. The Wang et al. (2015) reference. I am asking Reviewer 1 for the proper reference so that the authors can track that paper down and, if suitable, incorporate relevant information (e.g., microsat markers).

[Quoted text hidden]

Vivienne Litzke <vivienne.litzke@gmail.com>

Royal Society Open Science - Decision on Manuscript ID RSOS-200419.R1

Joe Hoffman <j_i_hoffman@hotmail.com>

Wed, Jun 17, 2020 at 1:02 PM

To: Open Science <Open.Science@royalsociety.org>, Vivienne Litzke <vivienne.litzke@gmail.com>, Rebecca Nagel <rebecca.nagel@uni-bielefeld.de>

Dear Anita

Many thanks for this. Vivienne - would you mind uploading the final files, please? Or if nothing needs to change, surely the previous files you uploaded should suffice?

Best wishes

Joe

Prof. Joseph Hoffman
Department of Animal Behaviour
University of Bielefeld
Postfach 100131
33501 Bielefeld
Germany
+49 (0)521 1062711
<http://www.thehoffmanlab.com>

From: Open Science <Open.Science@royalsociety.org>**Sent:** 17 June 2020 11:00**To:** Joe Hoffman <j_i_hoffman@hotmail.com>**Subject:** RE: Royal Society Open Science - Decision on Manuscript ID RSOS-200419.R1

Dear Joe et al.,

Thank you again for your message.

I've now contacted the Associate Editor, and they're happy for you to proceed with your revision without this reference. Please submit your revised manuscript as soon as conveniently possible.

Please do let us know if you require any additional time!

Best regards,

Anita Kristiansen
Editorial Coordinator

T +44 20 7451 2633
Royal Society Open Science

The Royal Society
[6-9 Carlton House Terrace](https://royalsociety.org/journals/)
London SW1Y 5AG

<https://royalsociety.org/journals/>

Registered Charity No. 207043

[Quoted text hidden]

This email is sent on behalf of The Royal Society, 6-9 Carlton House Terrace, London SW1Y 5AG, United Kingdom.

The contents of this email and any attachments are intended for the confidential use of the named recipient(s) only. They may be legally privileged and should not be communicated to or relied upon by any person without our express written consent. If you are not an addressee (or you have received this mail in error) please notify us immediately by email to: it.admin@royalsociety.org, and confirm the deletion of this email and attachments immediately. You should carry out your own virus check before opening any attachment. The Royal Society accepts no liability for any loss or damage which may be caused by software viruses or interception or interruption of this email.

Please see our privacy policy for details of how any personal data we collect from you, or that you provide to us, will be processed by us.

Registered charity no. 207043

The views or opinions are solely those of the author of this email, and do not represent those of The Royal Society unless specifically stated.